# Assessing Concentration Changes of Odorant Compounds in the Thermal-Mechanical Drying Phase of Sediment-Like Wastes from Olive Oil Extraction

**Diógenes Hernández** [1,2,3,*] **, Héctor Quinteros-Lama** [2] **, Claudio Tenreiro** [2] **and David Gabriel** [3]

[1]    Institute of Chemistry of Natural Resources, University of Talca, Box 747 Talca, Chile
[2]    School of Engineering, University of Talca. Merced 437, Curicó 3341717, Chile;
        hquinteros@me.com (H.Q.-L.); ctenreiro@utalca.cl (C.T.)
[3]    GENOCOV Research Group. Department of Chemical, Biological and Environmental Engineering,
        Universitat Autònoma de Barcelona, 08193 Bellaterra, Spain; david.gabriel@uab.cat
*    Correspondence: dhernandez@utalca.cl; Tel.: +56-75201764



**Featured Application: Authors are encouraged to provide a concise description of the specific application or a potential application of the work. This section is not mandatory.**

**Abstract:** In the industrial production of olive oil, both solid wastes and those produced from their incineration are a serious environmental problem since only 20% $w/w$ of the fruit becomes oil and the rest is waste, mainly orujo and alperujo. A key aspect to transforming these wastes into an important source of energy such as pellets is to recognize the most appropriate time of the year for waste drying, with the objective of minimizing the environmental impact of the volatile compounds contained in the waste. In this work, the emissions produced during thermal-mechanical drying were studied throughout a period of six months of waste storage in which alperujo and orujo were stored in open containers under uncontrolled environmental conditions. The studied emissions were produced when both wastes were dried in a pilot rotary drying trommel at 450 °C to reduce their initial humidity of around 70–80% $w/w$ to 10–15% $w/w$. Results indicated that when the storage time of the wastes in the uncontrolled environments increased, the emission of odorant compounds during drying also increased as a consequence of the biological and chemical processes occurring in the containers. The main odorant VOCs were quantified monthly for six months at the outlet of the drying trommel. It was determined that the drying of this type of waste could be carried out properly until the third month of storage. Afterwards, the concentration of most VOCs produced widely exceeded the odor thresholds of selected compounds.

**Keywords:** agroindustrial waste; alperujo and orujo; rotary dryer emissions; volatile organic compounds; biomass drying

## 1. Introduction

Between 2016 and 2017, the production of olive oil reached a global amount of 2.9 million tons. The European Union occupied 62% of the production corresponding to 1.8 million tons. This production was concentrated in Spain, Italy and Greece [1]. Despite the fact that Chile only represents 0.5% of the worldwide production, a progressive increase in recent years has been observed, especially by the incorporation of Chilean products in the national and international market in countries such as China, USA and Australia, reaching today about 1700 million consumers. These figures have generated a sustained growth in the amount of land planted with olive trees in Chile, increasing by 32% compared to 2004 [2].

Since 20% $w/w$ of olive oil is obtained in the extraction process and about 80% $w/w$ corresponds to waste, the sustained increment in the production has led to higher waste generation and to a significant increase of environmental impacts such as odor nuisance and groundwater contamination [3]. Olive mill solid wastes are mainly alperujo and orujo. It is important to note that olive oil extraction can be carried out in three different ways in relation to the types and characteristics of the waste generated and the costs of the process: (a) traditional pressing, which generates oil, water and olive cake; (b) two phases, in which oil and alperujo are obtained; (c) three phases, in which oil, alpechin and orujo are obtained (Kapellakis et al., 2008).

Alperujo is the residue obtained from the two-phase process, which is the most usual in Chile and worldwide because it consumes less water (Espionola, 1997), while orujo is the residue of the three-phase process. Both are an aqueous mixture rich in organic matter and other derivatives [4]. Studies conducted by Christoforou [5] describe olive-mill solid wastes as an important source of energy given that their calorific value is between 15.6 MJ kg$^{-1}$ to 19.8 MJ kg$^{-1}$ in dry basis. However, one of their disadvantages is the high moisture content, which lies between 60%–70% $w/w$, respectively [3,6]. This implies that these residues should be dried before any further use to reach a moisture content of around 8% $w/w$ in a dry basis. This process has been carried out for many years to obtain a product with added value, such as in the case of oils extracted with solvents [7]. The most used systems for this purpose are rotary dryers [8]. Such drying systems produce evaporation of volatile organic compounds (VOCs) when the temperature in the chamber increases; they also produce condensable compounds and particle emissions as a consequence of volatilization, steam production and thermal destruction [7,9]. Additionally, significant risks of producing polycyclic aromatic hydrocarbons due to the adherence of the residues to the drum walls exist [8]. Emissions produced depend on waste composition, which varies over time during waste storage due to the effect of environmental variables (temperature, rainfall and humidity) on the chemical and biological processes occurring in the storage reservoirs [10]. In addition to the fact that emissions from thermal-mechanical drying of such waste have been poorly addressed in the literature, a significant variability of their composition is expected considering the influence of the storage time in the release of VOCs from open reservoirs. Such variability has not been addressed previously in literature to the authors' knowledge.

Fagernäs et al. [9] determined that when drying temperatures are below 100 °C, monoterpenes and sesquiterpenes are produced during biomass drying. Conversely, when temperatures are above 100 °C, compounds such as fatty acids, resin acids and higher terpenes are produced, which are responsible for haze formation around these kinds of systems due to air cooling, causing odor and visual discomfort. On the one hand, Fagernäs et al. [9] determined that other compounds released during drying processes are alcohols, organic acids and aldehydes, which are the product of thermal destruction of hemicellulosic materials when temperatures exceed 150 °C. In contrast, a study conducted by Jauhiainen et al. [11] evaluated gaseous emissions in the pyrolysis and combustion processes in a tubular laboratory batch reactor of olive oil solid waste at different temperatures from 750 to 1050 °C. This study showed that a great abundance of compounds is generated and that their concentrations and type of compounds increase along with temperature. The most abundant molecules found were methane, benzene, ethene, ethane, propane, propene, butane, butene, butadiene and toluene.

The aim of this work was to characterize and quantify the VOCs produced when the alperujo and orujo were dried using a pilot-scale rotary dryer (trommel). Both residues were stored outdoors and under uncontrolled environmental conditions for six months, simulating the actual industrial conditions, to evaluate the impact of the storage time on the dryer emissions. Thus, the initial humidity for each residue fluctuated between 50–80% $w/w$ and was reduced to around 10–15% $w/w$ in order to be suitable for pelletization.

## 2. Materials and Methods

### 2.1. Identification of the Samples

In June 2017, six samples from two different mills were obtained, three for alperujo and three for orujo. Samples were taken from the Maule Region, Chile. Samples of orujo and alperujo were taken from the oil mills from the only waste-outlet duct and deposited in 200 L polypropylene containers. The procedure for obtaining the samples as well as the geographical location of the oil milling sites are described in Hernández et al. [10]. Each sample was identified and standardized to a mass of 200 kg and moved and stored under uncontrolled environmental conditions outdoors in open containers in the Campus Curicó, University of Talca, Curicó, Chile.

### 2.2. Environmental Conditions

Provided that the containers were located outdoors and in order to evaluate the effect of environmental conditions, a daily measurement for temperature, humidity, wind speed and amount of rainfall was carried out during the entire experimental period of six months, which was averaged on a monthly basis (Table S1 in the supplementary material). These parameters were studied since relevant information can be obtained regarding the changes that can occur in the odorant compounds due to the exposure time and the climate variability related to the rainfall and the temperatures as the year progresses from winter to spring. In any case, the influence of the environmental conditions on the samples in the study was found to be similar to that already reported by Hernández et al. [10]. This study was performed for samples of the same type of wastes and exposed to environmental conditions in the same location but performed in the year prior to this current study, concluding that the key factor in the change of the properties of the waste is the environmental temperature followed by the rain to a lesser extent.

Representative samples were extracted every month for a six-month period from each container containing alperujo and orujo waste as described elsewhere [10]. Every month, subsamples from different parts of the container of each residue were mixed independently in a 50 L polypropylene container and manually stirred to obtain a final homogeneous sample. These mixtures of orujo and alperujo were separated every month as follows: 5.0 kg frozen at $-18\,^{\circ}$C stored as control samples; 1.0 kg for laboratory tests in triplicate, using standardized methods according to DIN [12] and AOAC International [13] and 15.0 kg of orujo and alperujo samples that were divided into three subsamples of 5.0 kg each to be dried in the drying trommel in triplicate. Table 1 shows the main characteristics of the physical-chemical parameters analyzed of each waste prior to its drying phase.

**Table 1.** Results of the physical-chemical analysis for samples of alperujo and orujo corresponding to the initial conditions before processing in the drying trommel over time. Deviations correspond to the standard deviation of triplicates.

| Month | Moisture Content (% wb) ** Oven Method—AOAC 945.15 | | Ashes (% wb) * Muffle Method—AOAC 940.26 | | Measuring Heating Value (MJ kg$^{-1}$) * Norm DIN Serie 51.900 | |
|---|---|---|---|---|---|---|
| | Alperujo | Orujo | Alperujo | Orujo | Alperujo | Orujo |
| Jul | 77.7 ± 0.1 | 79.5 ± 0.2 | 2.4 ± 0.1 | 3.2 ± 0.1 | 22.3 ± 0.1 | 22.4 ± 0.2 |
| Aug | 77.9 ± 0.5 | 80.2 ± 0.3 | 2.2 ± 0.1 | 3.1 ± 0.1 | 22.1 ± 0.2 | 22.4 ± 0.1 |
| Sept | 80.0 ± 0.5 | 81.8 ± 0.4 | 1.9 ± 0.2 | 2.9 ± 0.1 | 22.4 ± 0.1 | 22.5 ± 0.1 |
| Oct | 73.1 ± 0.2 | 77.4 ± 0.2 | 1.6 ± 0.1 | 2.7 ± 0.1 | 22.2 ± 0.1 | 22.4 ± 0.2 |
| Nov | 60.5 ± 0.4 | 63.4 ± 0.2 | 1.6 ± 0.1 | 2.2 ± 0.1 | 22.1 ± 0.1 | 22.6 ± 0.2 |
| Dec | 53.0 ± 1.0 | 55.4 ± 0.4 | 1.2 ± 0.2 | 2.0 ± 0.1 | 22.3 ± 0.1 | 22.7 ± 0.2 |

* Dry basis; ** wet basis.

Each month, alperujo and orujo samples were introduced in triplicate in a pilot-scale rotary drying trommel shown in Figure 1. The drying trommel consisted mainly of a turbo-heater (2) of propane-butane gas (model SG30) with a direct flame that provides the caloric energy to the system. The entry feeding hopper (1) for the wet material (alperujo and orujo) had a height of 50 cm, the diameter in the upper part was 50 cm and the diameter in the lower part was 20 cm. The concentric drum (10) had a length of 2500 cm and 80 cm in diameter. In addition, paddles were built inside the drum to allow samples to be homogenized. Finally, the trommel had two orifices, one for the exit of the dry biomass (11) and the other for the exit of the exhaust gases and particulate matter (8). The exhaust gases and particulate matter passed through a separation cyclone (7) to extract fine particles, while the hot gases were released through an outlet pipe where gas samples were taken for Volatile Organic Compounds (VOCs) analysis.

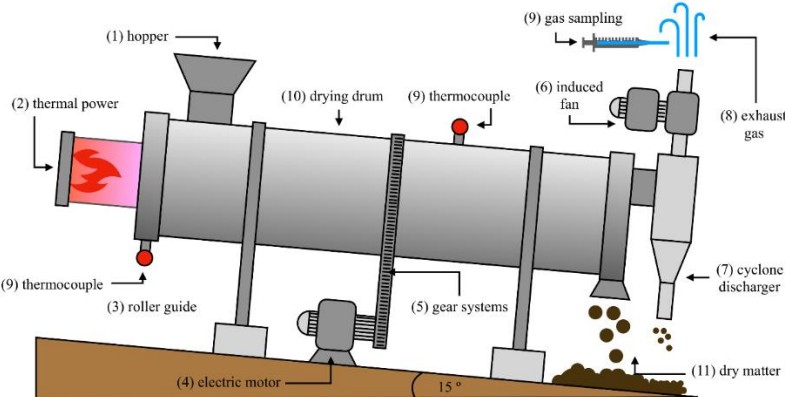

**Figure 1.** Biomass drying trommel. (1) hopper; (2) thermal power; (3) roller guide; (4) motor; (5) gear systems; (6) induced fan; (7) cyclone discharger; (8) exhaust gas; (9) thermocouple; (10) drying drum; (11) dry biomass output.

The working conditions of the drying trommel were selected based on those normally used in the olive oil production industry. A working temperature in the concentric drum of $420 \pm 10\,^\circ$C was selected and controlled using two thermocouples (PT100), one for the entry of hot gases into the system and the other inside the drum of the drying trommel. The drum rotation speed of the system was programmed at 5.0 RPM and an inclination of $15^\circ$ in order to achieve a better displacement of the material inside the drum and achieve a homogeneous drying in a total time of approximately 15 min. The final outlet moisture of the alperujo and orujo was between 10–15% $w/w$ on dry basis.

For each sample of alperujo and orujo, three tests per sample were carried out allowing to evaluate the outlet emissions and waste characteristics after the drying process (Table S2, supplementary material). In all cases, the average humidity of the material after the tests fluctuated between 11–13% $w/w$ on dry basis, while the average for ash content and the measuring heating value were 1.5–3.0% $w/w$ and 22.3 MJ kg$^{-1}$, respectively, for both alperujo and orujo.

*2.3. Sampling of VOCs and Conditions of Thermal Desorption-Gas Chromatography/Mass Spectrometry (TD-GC/MS)*

At a distance of 50 cm from the outlet pipe of the exhaust gases of the drying trommel, three samples were taken for each test at different times (5 min, 10 min and 15 min) at an average outlet temperature of 150 $^\circ$C. In order to extract the gas samples, a suction pump (Markes Easy VOC model LP-1200, Germany) was used, extracting an amount of 100 mL through an absorbent tube (Markes C2-BAXX-5315 odor/sulfur. C6/7-C30, thiols and mercaptans, Germany). Once the gases were adsorbed, VOCs desorption was carried out using a cold trap of thermal desorption (Markes model Unity-xr, Germany). The gases were extracted in Split mode, driven with helium for 1 min to the hot trap programmed at 300 $^\circ$C and then cooled to 20 $^\circ$C in the cold trap to finally be

heated to 300 °C for 5 min. VOCs were transferred by means of a transfer line heated at 200 °C to one column (RESTEK-Rtx-5MS, PA, USA. w/integra-guard Crossbond 5% diphenyl–95% dimethyl polysiloxane. 30 m, 0.25 mmID, 0.25 μm df) installed in a GC/MS (Thermo Fisher Scientific, model Trace 1300/ISQELTL, MA, USA). The working conditions of the GC for the oven were in Split mode with a working temperature between 40 °C and 220 °C. Flow: 1.2 mL/min; Split Ratio: 10 °C/min, the transfer line temperature of the MS detector was 200 °C while the ion source temperature was set at 250 °C. The qualitative identification of VOCs was carried out using the Chromeleon 7.2 software package, which is compatible with the NIST library [14], using retention times observed in the chromatograms.

### 2.4. Reagents

All the standards used for the quantification of VOC concentrations in the TD/GC-MS corresponded to substances with purities greater than 95%. The calibration curves were constructed using four different mixtures based on methanol with *n*-hexane, *n*-octane, *n*-nonane and D3710-95 from Restek, which is composed of 16 compounds. Other standards were purchased individually from Merck, Sigma-Aldrich and Lancaster Synthesis. The concentration of each standard is summarized in Table 2.

**Table 2.** Concentration of the standard for the quantification of VOC concentrations.

| Based Compound | Sample | Standard 1 | Standard 2 | Standard 3 |
| --- | --- | --- | --- | --- |
| Methanol | *n*-Hexane | 0.5 ppm$_v$ | 1.5 ppm$_v$ | 3.0 ppm$_v$ |
| Methanol | *n*-Octane | 0.5 ppm$_v$ | 1.5 ppm$_v$ | 3.0 ppm$_v$ |
| Methanol | *n*-Nonane | 0.5 ppm$_v$ | 2.0 ppm$_v$ | 5.0 ppm$_v$ |
| Methanol | D3710-95 | 0.5 ppm$_v$ | 1.0 ppm$_v$ | 2.5 ppm$_v$ |

## 3. Results and Discussion

Among all the samples extracted from the exhaust gases during alperujo and orujo drying and later analyzed by TD-GC/MS, more than 600 compounds were identified. However, a large list of them only appeared a few times or in small abundances, well below their odor threshold. In addition, the reliability of some identified compounds was below 98% coincidence with the NIST library [14]. Thus, inventory of all the compounds found over time was reduced to these shown in Table 3, in which VOCs were classified by families, including the months in which each VOC was detected during the course of the study. Only those compounds that were found in more than 25% of the 108 output samples were considered representative in terms of emissions, following the criteria described by Dorado et al. [15] for compost maturation emissions in a wastewater treatment plant. The compounds considered in the analysis have a ratio in the areas in comparison to the total area of the sample with an average of 63.62% for July to 92.95% for December, monotonously increasing as time passes.

**Table 3.** Selected VOCs detected during the drying process for alperujo and orujo samples.

| Family | Name | Months (1–6) Alperujo | Orujo | Odor Threshold ppm$_v$ | Reference |
|---|---|---|---|---|---|
| **Aldehydes** | 2-Furancarboxaldehyde, 5-methyl- | 1,3,4 | 1 | 3.0 | Sung et al. [16] |
| | 3-Cyclopentene-1-acetaldehyde, 2-oxo- | 1,3 | 1–3 | | |
| | 9-Octadecenal | 3,4 | | >1.0 | Caprino et al. [17] |
| | Benzaldehyde | 1–6 | 1–6 | 0.35–3.5 | Buttery et al. [18] |
| | Furfural | 1–6 | 1–6 | 3.0–23.0 | Buttery et al. [18] |
| | Hexanal | 1–6 | 1–6 | 0.28 | Nagata [19] |
| | Methyl glyoxal | 3,4 | 3–5 | | |
| | Nonanal | 1–6 | | 0.34 | Nagata [19] |
| | Octanal | 1–6 | | 0.01 | Nagata [19] |
| **Amides** | Propanamide, 2-hydroxy- | 2–6 | 1–6 | | |
| **Amines** | Pyridine, 3-ethyl- | 2–3 | 2–3 | | |
| **Phenolic alcohols** | | | | | |
| | 2-Methoxy-4-vinylphenol | 2–3 | 1–6 | 0.003 | Nagata [19] |
| | 3-tert-Butyl-4-hydroxyanisole | 2–6 | 2–3 | | |
| | Phenol | 1–6 | 1–6 | 0.0056 | Nagata [19] |
| | Phenol, 2-methoxy-4-(1-propenyl)- | 2–5 | 1–6 | | |
| | Phenol, 2,6-dimethoxy- | 1–2 | 1 | | |
| | Phenol, 4-ethyl-2-methoxy- | 1–5 | 1–6 | | |
| | 5-tert-Butylpyrogallol | 2–3 | 2-3 | | |
| **Aliphatic alcohols** | 1-Dodecanol, 3,7,11-trimethyl | | 1–3 | | |
| | 1-Dodecanol, 3,7,11-trimethyl- | 2–5 | 2–5 | | |
| | 1-Hexadecanol, 2-methyl- | 1 | 1–2 | | |
| | 3-Nonen-1-ol, (E)- | | 2–3 | | |
| | 2-Furanmethanol | 1–6 | 1-6 | 8.0 | Montgomery [20] |
| **Aromatic HCs** | Benzene, 1-azido-3-methyl- | 2–3 | 2 | | |
| | Benzene, 1,3-dimethyl- | 2–5 | 2 | | |
| | Toluene | 1–6 | 1–6 | 0.33 | Nagata [19] |
| **Esters** | 10-Octadecenoic acid, methyl ester | 2–3 | | | |
| | Hexanoic acid, 2-phenylethyl ester | 2–6 | | | |
| **Carboxylic acids** | Acetic acid | 1–6 | 1–6 | 0.0060 | Nagata [19] |
| | Propanoic acid | 4–5 | | 0.0057 | Nagata [19] |
| | 9-Hexadecenoic acid | 1–6 | | | |
| **Aliphatic HCs** | 1-Decene | 3–6 | 5–6 | | |
| | 2-Octene | | 4–5 | 99.808 | http://www.odourthreshold.com [21] |
| | 8-Heptadecene | 5 | 3–6 | | |
| | Heptane | | 1–6 | | |
| | Hexane | 1–6 | 1–6 | 1.5 | Nagata [19] |
| | Nonane | 1–6 | 1–6 | 2.2 | Nagata [19] |
| | Octane | 1–6 | 1–6 | 1.7 | Nagata [19] |
| | Tetradecane, 2,6,10-trimethyl- | 3–6 | 1–6 | | |
| | 1,4-Pentadiene | 2–3 | 1–2 | | |
| **Ketones** | 1,2-Cyclopentanedione, 3-methyl- | 5–3 | 2 | | |
| | 2-Cyclopenten-1-one, 2-hydroxy-3-methyl- | 1–6 | 4–5 | | |
| | 2-Cyclopenten-1-one, 2-methyl- | 3–6 | 1–6 | | |
| | 2-Propanone, 1-(acetyloxy)- | 4–5 | 1–6 | | |
| | 2-Propanone, 1-hydroxy- | 2–6 | 5 | | |
| **Ethers** | Octane, 1-methoxy- | 1–2 | 6 | | |

Table 3 shows that there are a significant number of compounds repeated over time for the samples of alperujo and orujo. Abundance is shown as 1–6, indicating that these compounds appeared in all the sampling events throughout the 6 months of experimentation, while those that appeared as a single number, for example 1, are those that only appeared in the first month (June) of experimentation. The compounds found more frequently (all months from 1 to 6) in both alperujo and orujo corresponded mainly to aldehydes (benzaldehyde, furfural and nonanal), alcohols (phenol, 2-furanmethanol), carboxylic acids (acetic acid) and hydrocarbons (toluene, hexane, octane and nonane). This leads to consider such molecules as target compounds for further treatment of emissions from the drying process. Consequently, these compounds were later selected for further quantification. Interestingly, Fagernäs et al. [9] showed that these types of molecules appear mostly when the temperature exceeds 150 °C because the process of thermal destruction of the wood hemicellulose begins. In addition, Dalai et al. [22] demonstrated that when the temperature in the rotary drying system increased from 60 °C to 250 °C in a time of 45 min for the drying of alfalfa, the presence of VOCs increased, especially hydrocarbons, alcohols, aldehydes and ketones. Thus, emissions are increased when the

temperature in the drying system is increased, consequently increasing their environmental impact. Results were in agreement with those found by Jauhiainen et al. [11], in which the emissions produced in the combustion of olive oil solid waste were mainly hydrocarbons, such as ethane, propane, hexane, toluene, benzene as well as pyridinic derivatives and polycyclic aromatic hydrocarbons. This demonstrates that the most frequent molecules in Table 3 were representative of the emissions during the thermal-mechanical drying of these wastes. It is worth mentioning that a limited number of references were found regarding the odor threshold concentration for the compounds emitted from alperujo and orujo drying, highlighting that further research is needed to complete that gap. Fortunately, odor thresholds of those compounds found as representative of alperujo and orujo emissions during waste drying have been previously reported (Table 3).

In addition to the factors that influenced the drying process, VOCs found in the exhaust gases were also related to the prior storage stage as reported previously. Firstly, Hernández et al. [3] and Rodríguez et al. [23] showed that there are still a number of important oils (4.0% and 5.7%) in olive oil solid waste prior to drying. Secondly, Morales et al. [24] demonstrated that the main VOCs released from sour olive oils were hexanal, octane and acetic acid among other odorant compounds, which were identified as responsible for unpleasant odors when they exceeded their odor threshold concentration. Additionally, Hernández et al. [10] found that alperujo and orujo wastes generate VOCs, especially families of aldehydes and carboxylic acids when they are stored in open reservoirs in uncontrolled spaces. Moreover, the emissions' composition was influenced by the storage time and the environmental conditions.

Figure 2 shows the distribution over time of the VOCs families considered in Table 3 throughout the six months of study. Alcohols, carboxylic acids and aldehydes were those families that presented the highest percentage of compounds corresponding to an average of 36%, 27% and 15%, respectively, for alperujo; and 41%, 21% and 10%, respectively, for orujo. In a previous study, Hernández et al. [10] determined that alcohols, carboxilic acids and aldehydes were also the predominant compounds in the emissions both from alperujo and orujo stored in open reservoirs due to a progressive decrease in the concentration of fatty acid methyl esters and alkenes in the emissions generated by biological and physical-chemical processes. Despite that most of the families showed no clear variability along the monitored period, still some families of VOCs showed a trend towards an increase or decrease over time. An increasing trend was observed for aliphatic and aromatic HCs in alperujo (Figure 2a) and for esters in orujo (Figure 2b).

When results are analyzed based on the humidity of the samples (Table 1) and the rain precipitations (Table S1, supplementary material), there were no significant changes in the abundance of VOC families for the waste drying process. The endothermic drying of waste has the purpose of stabilizing the waste, reducing its weight and its volume, the latter as a consequence of water loss by evaporation by the addition of heat. In addition, Castells et al. [25] have shown that water when evaporated transports most of the VOCs, which are released to the atmosphere.

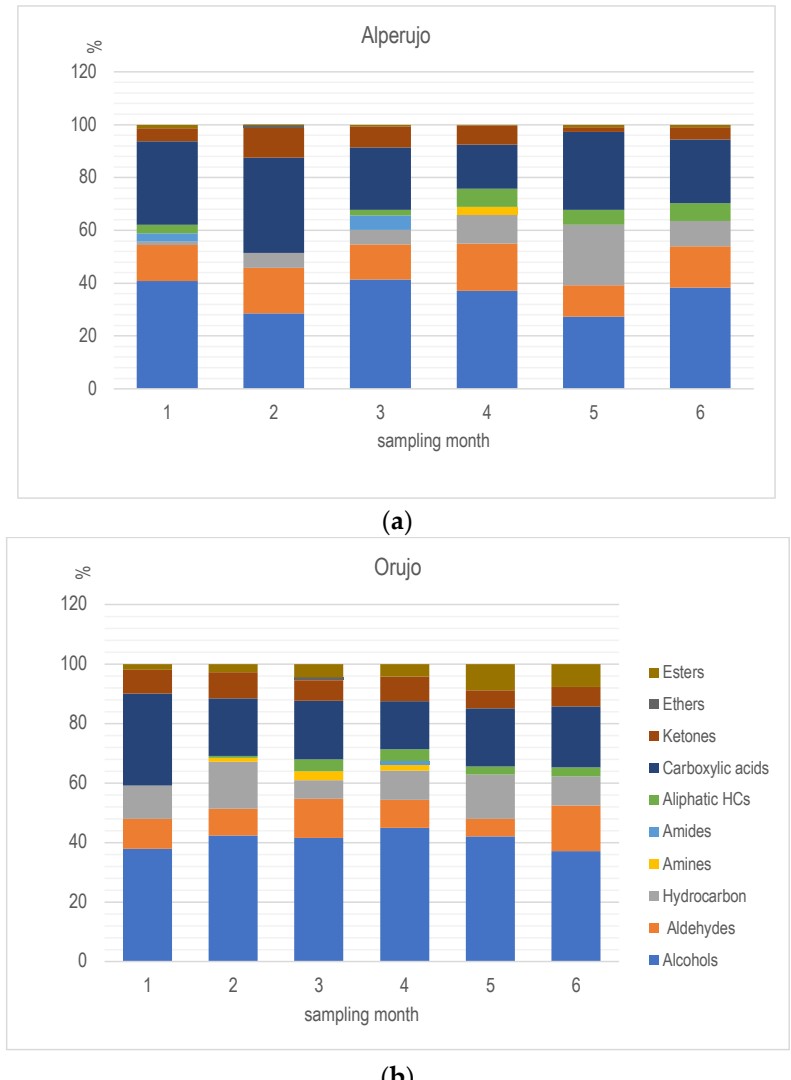

**Figure 2.** Percentage distribution of VOCs by family in (**a**) alperujo and (**b**) orujo in the six-month experimentation period.

Dexter et al. [26] described that VOCs have a wide range of evaporation temperatures. Most of them are evaporated at lower temperatures than water and their vaporization enthalpies are of the order of magnitude of water. Therefore, when water and VOCs are released from the waste in the form of vapor and contained within the mass of air circulating through the drying trommel, they are removed from the system by depleting varying portions of the air mass charged with steam through the discharge and replacement of this exhaust air with equivalent amounts of fresh air containing less humidity. Tables 4 and 5 show the normal boiling temperatures of each of the key odorant compounds found, which are lower than the operating temperature in the dryer.

**Table 4.** Normal boiling temperature, vapor pressure at 150 °C and the concentration of target VOCs in ppm$_v$ in exhaust gases during alperujo drying.

| Family | Name | Boiling Temperature | Vapor Pressure | Jul | Aug | Sep | Oct | Nov | Dec | |
|---|---|---|---|---|---|---|---|---|---|---|
| | | °C | bar | 1 | 2 | 3 | 4 | 5 | 6 | average |
| Alcohols | 2-Furanmethanol | 170.10 * | 0.45 | 3.0 ± 0.1 | 3.1 ± 0.1 | 3.5 ± 0.1 | 4.0 ± 0.1 | 4.2 ± 0.1 | 5.2 ± 0.2 | 3.8 ± 0.5 |
| Phenol | 181.93 * | 0.40 | 0.1 ± 0.1 | 0.2 ± 0.1 | 0.3 ± 0.1 | 0.4 ± 0.1 | 0.9 ± 0.2 | 1.0 ± 0.1 | 0.5 ± 0.4 | |
| Aldehydes | Benzaldehyde | 178.66 * | 0.46 | 3.5 ± 0.0 | 4.0 ± 0.1 | 4.2 ± 0.1 | 4.8 ± 0.2 | 4.9 ± 0.1 | 5.1 ± 0.1 | 4.4 ± 0.6 |
| Furfural | 161.55 ** | 0.73 | 6.3 ± 0.1 | 7.5 ± 0.1 | 7.8 ± 0.1 | 8.3 ± 0.1 | 8.9 ± 0.0 | 10.5 ± 0.0 | 8.2 ± 1.4 | |
| Hexanal | 128.14 * | 1.80 | 0.1 ± 0.1 | 0.5 ± 0.0 | 0.8 ± 0.0 | 1.2 ± 0.0 | 1.3 ± 0.1 | 1.5 ± 0.0 | 0.9 ± 0.5 | |
| Nonanal | 194.93 * | 0.29 | 0.3 ± 0.0 | 0.6 ± 0.1 | 0.9 ± 0.1 | 1.5 ± 0.0 | 1.8 ± 0.1 | 2.1 ± 0.0 | 1.2 ± 0.7 | |
| Octanal | 174.20 * | 0.53 | 0.1 ± 0.0 | 0.1 ± 0.1 | 0.5 ± 0.1 | 0.9 ± 0.0 | 1.1 ± 0.2 | 1.2 ± 0.1 | 0.7 ± 0.5 | |
| Aromatic HCs | Toluene | 110.68 * | 2.75 | 0.5 ± 0.1 | 0.7 ± 0.2 | 0.7 ± 0.1 | 0.8 ± 0.1 | 0.9 ± 0.1 | 1.2 ± 0.1 | 0.8 ± 0.2 |
| Carboxylic acids | Acetic acid | 118.01 * | 2.49 | 2.8 ± 0.1 | 3.5 ± 0.1 | 2.9 ± 0.1 | 3.5 ± 0.1 | 4.2 ± 0.0 | 4.9 ± 0.1 | 3.6 ± 0.8 |
| Aliphatic HCs | Hexane | 68.73 * | 7.41 | 0.9 ± 0.0 | 1.9 ± 0.1 | 2.9 ± 0.1 | 2.8 ± 0.0 | 2.5 ± 0.1 | 1.5 ± 0.2 | 2.1 ± 0.8 |
| Octane | 125.69 * | 1.90 | 3.2 ± 0.0 | 3.2 ± 0.1 | 3.5 ± 0.2 | 2.1 ± 0.0 | 1.0 ± 0.1 | 0.5 ± 0.1 | 2.3 ± 1.3 | |
| Nonane | 150.66 * | 0.99 | 2.1 ± 0.1 | 2.9 ± 0.1 | 3.0 ± 0.1 | 1.5 ± 0.1 | 1.9 ± 0.1 | 2.0 ± 0.1 | 2.2 ± 0.6 | |

* DIPPR [27] ** NIST [14].

**Table 5.** Normal boiling temperature, vapor pressure at 150 °C and the concentration of target VOCs in ppm$_v$ in exhaust gases during orujo drying.

| Family | Name | Boiling Temperature | Vapor Pressure | Jul | Aug | Sep | Oct | Nov | Dec | |
|---|---|---|---|---|---|---|---|---|---|---|
| | | °C | bar | 1 | 2 | 3 | 4 | 5 | 6 | average |
| Alcohols | 2-Furanmethanol | 170.10 * | 0.45 | 3.1 ± 0.1 | 3.9 ± 0.2 | 4.2 ± 0.2 | 4.2 ± 0.1 | 4.9 ± 0.0 | 5.7 ± 0.2 | 4.3 ± 0.9 |
| Phenol | 181.93 * | 0.40 | 0.1 ± 0.1 | 0.1 ± 0.1 | 0.8 ± 0.2 | 0.7 ± 0.1 | 1.2 ± 0.1 | 1.8 ± 0.2 | 0.8 ± 0.7 | |
| Aldehydes | Benzaldehyde | 178.66 * | 0.46 | 3.1 ± 0.2 | 3.2 ± 0.1 | 3.8 ± 0.1 | 4.3 ± 0.1 | 4.9 ± 0.0 | 4.9 ± 0.1 | 4.0 ± 0.8 |
| Furfural | 161.55 ** | 0.73 | 3.9 ± 0.1 | 4.3 ± 0.1 | 5.8 ± 0.1 | 5.5 ± 0.2 | 6.9 ± 0.1 | 8.6 ± 0.0 | 5.8 ± 1.7 | |
| Hexanal | 128.14 * | 1.80 | 1.7 ± 0.0 | 1.1 ± 0.1 | 1.3 ± 0.1 | 2.1 ± 0.2 | 2.2 ± 0.1 | 2.3 ± 0.0 | 1.8 ± 0.5 | |
| Nonanal | 194.93 * | 0.29 | 0.2 ± 0.0 | 0.5 ± 0.0 | 0.9 ± 0.1 | 0.9 ± 0.1 | 1.1 ± 0.0 | 1.5 ± 0.0 | 0.9 ± 0.5 | |
| Octanal | 174.20 * | 0.53 | 0.2 ± 0.1 | 0.2 ± 0.0 | 0.8 ± 0.1 | 1.2 ± 0.2 | 1.8 ± 0.2 | 2.1 ± 0.1 | 1.1 ± 0.8 | |
| Aromatic HCs | Toluene | 110.68 * | 2.75 | 0.8 ± 0.1 | 0.8 ± 0.2 | 1.2 ± 0.2 | 1.3 ± 0.1 | 1.7 ± 0.2 | 2.1 ± 0.1 | 1.3 ± 0.5 |
| Carboxylic acids | Acetic acid | 118.01 * | 2.49 | 2.4 ± 0.1 | 3.2 ± 0.1 | 3.8 ± 0.1 | 4.1 ± 0.0 | 4.5 ± 0.1 | 5.2 ± 0.1 | 3.9 ± 1.0 |
| Aliphatic HCs | Hexane | 68.73 * | 7.41 | 2.1 ± 0.1 | 2.0 ± 0.1 | 2.1 ± 0.2 | 1.9 ± 0.1 | 1.2 ± 0.0 | 1.0 ± 0.2 | 1.7 ± 0.5 |
| Octane | 125.69 * | 1.90 | 1.2 ± 0.1 | 2.1 ± 0.1 | 2.2 ± 0.3 | 1.9 ± 0.1 | 0.5 ± 0.2 | 0.2 ± 0.0 | 1.4 ± 0.9 | |
| Nonane | 150.66 * | 0.99 | 2.4 ± 0.1 | 3.1 ± 0.1 | 3.4 ± 0.1 | 3.8 ± 0.0 | 4.2 ± 0.1 | 4.1 ± 0.1 | 3.5 ± 0.7 | |

* DIPPR [27] ** NIST [14].

In addition, Tables 4 and 5 show the quantification of VOCs that were considered as target compounds according to the results in Table 1. Concentrations were compared with the odor threshold concentrations also reported in Table 1.

Except for furfural and 2-furanmethanol, concentrations of all target compounds exceeded their odor threshold in all samples analyzed in both alperujo and orujo. A comparison of the odor thresholds in Table 1 with the concentrations of odorants in Tables 4 and 5 show that for alperujo and orujo, the average benzaldehyde concentration exceeded its odor threshold by 25.7% and 14.3%, respectively. Similarly, acid acetic exceeded their odor threshold in alperujo and orujo by 81,666% and 86,666%, respectively. Hexanal, nonanal, octanal, phenol, hexane and toluene concentrations exceeded their odor thresholds in alperujo by 221%, 253%, 6,900%, 9,900%, 40% and 142% and 543%, 165%, 10,900%, 15,900%, 13.3% and 294% in orujo, respectively. Considering that most of the compounds reported in Tables 4 and 5 are odorants that produce unpleasant odors when they are emitted to the environment [24], in practice, emissions from exhaust gases from oil mill waste drying need further treatment before their discharge to the environment to avoid a bothersome impact on the surrounding areas of the industrial facilities.

Figure 3 graphically shows the variability of odorant concentrations quantified over time. This tendency to increase or decrease can be directly associated with the physical-chemical and biological changes that occurred in the samples over time given their organic composition and environmental conditions such as temperature and humidity, registered in Table 1.  Hernández

et al. [10] describes that talperujo and orujo, when stored under uncontrolled environmental conditions over time, generate VOCs that increase their concentration as a result of the chemical and biological transformations that occur in these residues.

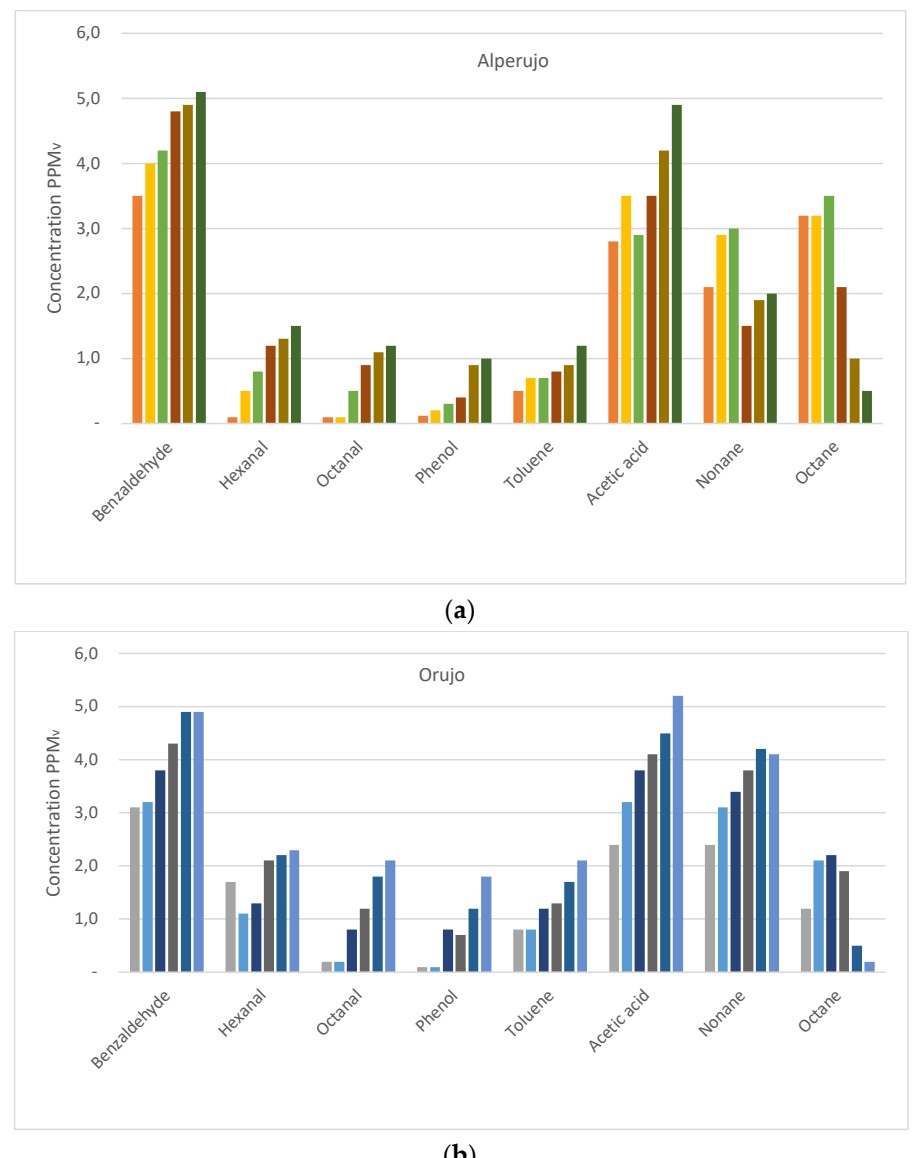

**Figure 3.** Monthly concentration of target VOCs found in (**a**) alperujo and (**b**) orujo drying emissions. In these graphs, bars of the groups represent each month where the experimentation was carried out between July to December from left to right.

Aldehydes, in particular hexanal and octanal, increased their concentration over time for both alperujo, as a direct influence of oxidative processes of the decomposition reactions of the hydroperoxides formed by the slow self-oxidation of oleic acid as well as the self-oxidation of lipoxygenase through the formation of hydroperoxide precursors 13-LOOH (linoleic acid hydroperoxide) [28,29]. It is important to note that the thermal degradation of the polysaccharide molecules has an impact on the compounds presented in the emissions. A number of compounds comes from the thermal drying itself but also an important group of compounds comes from the aforementioned degradation, in particular, acetic acid, acetaldehyde, furane and so forth.

Interestingly, octane was the only VOC that showed a decreasing trend both in alperujo and orujo. Kalua et al. [29] described that this compound is recognized as a marker of oxidation in oils since it

appears in high concentrations at the beginning of the process. In addition, Vichi et al. [28] indicated that octane is formed from hydroperoxide precursors 10-OOOH (oleic acid hydroperoxide) [28]. Thus, octane degradation can be a consequence of oxidative deterioration of oleic acid over time as described by Morales et al. [30]. In a 33-hour experiment, it was observed that the main volatile decomposition products found in oxidized olive oils were produced from monohydroperoxides of oleic, linoleic and linolenic unsaturated fatty acids that decreased their concentration over time, that is, in 98.2% at 0 h, 80.8% at 21 h and then 49.2% at 33 h. Considering that the odorants are emitted from open reservoirs during waste storage and that initial humidity of wastes is almost constant during the 3rd and4th months (Table 1), results indicate that a proper alternative to reduce the impact of exhaust gases from the oil mill waste drying from an emissions perspective is to process them as soon as they are produced to minimize the overall emission of odorants from the storage and drying. Finally, the compounds found in the outlet of the trommel may be considered similar to the original species before the drying [31] and hence similar to the species found by Hernandez et al. in previous work [3].

Odor concentration measured in the gaseous emissions of a treatment process can be complemented with the individual odor impact values, which can help in the identification of the major odor contributors found in a complex odorous gas mixture. Table 6 shows the odor impact value (OIV) per each VOC, which is obtained by dividing the average concentrations of each VOC quantified by its odor threshold according to literature data (Table 4).

**Table 6.** Odor impact value (OIV) for the individual VOCs based on average samples of alperujo and orujo over time.

| VOC Family | Compound | Odor Impact Value | |
|---|---|---|---|
| | | Alperujo | Orujo |
| Phenolic alcohols | Phenol | 89.29 | 142.86 |
| Aliphatic alcohols | 2-Furanmethanol | 0.48 | 0.54 |
| Aldehyde | Benzaldehyde | 1.26 | 1.14 |
| | Furfural | 0.36 | 0.25 |
| | Hexanal | 3.21 | 6.43 |
| | Nonanal | 3.53 | 2.65 |
| | Octanal | 70.00 | 110.00 |
| Aromatic | Toluene | 2.42 | 3.94 |
| | Acetic acid | 3.60 | 3.90 |
| Aliphatic HC | Hexane | 1.40 | 1.13 |
| | Nonane | 1.00 | 1.59 |
| | Octane | 1.35 | 0.82 |

Results show that aldehydes and aromatic compounds, mainly phenol and octanal, are the ones that present the highest OIV, implying that these families of compounds are the precursors of the odors when the alperujo and orujo are dried in the drying trommel over time. A study conducted by Morales et al. [24] shows that the high concentrations of aldehydes in olive oils (rancid, fusty, winey-vinegary, mustiness-humidity) are mainly produced by the oxidation of unsaturated fatty acids, while the presence of acids that appear in the final part of the oxidative process are the product of the oxidation of the previously formed aldehydes. In terms of emissions, results of the impact reduction from thermal-mechanical drying of alperujo and orujo point out that the development of treatment methods targeting particularly octanal, hexanal and phenol may be much more efficient in reducing bothersome impacts than common methods such as chemical scrubbers. Despite that further testing is warranted, such poorly soluble compounds emitted at high temperatures could probably be adsorbed efficiently in activated carbon beds.

## 4. Concluding Remarks

This work was devoted to showing the emissions of VOCs from samples of alperujo and orujo when they are stored over time and dried in a drying trommel. Results revealed that different odorant

VOCs at varying concentrations are generated over time, which are often emitted directly into the environment through a chimney. As a consequence, bothersome impacts in the surrounding vicinities will be generated. These compounds corresponded mainly to alcohols, aldehydes, aromatics, carboxylic acid and hydrocarbons, which were found in concentrations higher than their odor threshold. Results showed that from the third month onwards, the concentrations of VOCs classified as odorants were intensified. Thus, when alperujo and orujo wastes are stored over time and subjected to various environmental changes, there are ideal times to carry out the drying process without generating major consequences of odorant compounds both in the drying phase and storage phase, implying that the shorter the time the less environmental consequences are generated.

**Supplementary Materials:** The following are available online at http://www.mdpi.com/2076-3417/9/3/519/s1, Table S1: Average monthly weather conditions (www.wolframalpha), Table S2: Results of the physical-chemical analysis with standard deviation for samples of alperujo and orujo as exit conditions to the drying trommel in time.

**Author Contributions:** D.H. and D.G. conceived and designed the experiments; D.H. performed the experiments and the data analysis; C.T. contributed analysis tools; D.H. and H.Q.-L. wrote the paper.

**Funding:** This research was funded by the project PIEI Quim BIO, Universidad de Talca and FONDECYT, Chile (Project No. 11180103) (H. Quinteros-Lama).

**Acknowledgments:** Thanks to Agricola y Forestal don Rafael and Almazaras del Pacífico for providing us with the final residues of their processes. We also thank the Academic Writing Center in the Programa de Idiomas in the Universidad de Talca for their revision of the manuscript.

**Conflicts of Interest:** The authors declare no conflict of interest.

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
