# Peer review of "Assessing Concentration Changes of Odorant Compounds in the Thermal-Mechanical Drying Phase of Sediment-Like Wastes from Olive Oil Extraction"

_applsci, doi:10.3390/app9030519_

Reviewer 1 Report

The article is dealing is dealing with the thermal-chemical drying of sediment-like wastes from olive oil extraction and the VOC release. Therefore VOC emissions are monitored by TD-GC/MS during a 6 months period using a pilot rotary drying trommel unit.

The article is well-written but there are still some issues which need further comments and should be discussed in more detail

Lines 93 and 110: it is not clear what is meant by extraction. Is it a mechanical collection of the waste sample?

Line 93: What kind of plastic was used for the container?

Lines 138-144: the working temperature in the concentric drum is around 420°C. However what is the temperature of the biomass? As the gases at the outlet for sampling are at 150°C (line157) the temperature of the biomass inside the concentric drum should be above 200°C according to line 201 it should be 250°C). Therefore in my opinion the drying process is partly also to be considered as a kind of torrefaction which also induces volume reduction, higher energy density, more homogeneous and improved grindability/pelletization. In this case the moisture is assisting in the release of volatile low MW compounds.

Line 153: thermal desorption-gas chromatography/mass spectrometry

A thermogravimetric analysis (TGA) could mimic the drying process using the same heating program and the same air flow. The TG profiles could assist in obtaining a direct view (based on monitoring mass loss on) on the release of VOC. DTG profiles could assist in the evaluation of different steps/transitions during the thermal treatment.

Lines 172-176: more information is needed for the quantitative determination of VOCs? E.g. is there a spike of an empty adsorption tube with a standard mix? How many standards are used?

Line 179: in terms of peak area what was the sum of peak areas of the compounds which are reported in table 2 with respect to the total peak area?  

Table 12: 1,4-pentadiene is list in the class of amines? Within the class of alcohols I suggest to make a difference between phenolic and aliphatic alcohols since their contribution to odour is significant different.

Line 198-213: Because of the enhanced temperature it should explained that we have to consider different processes dealing with the release of volatiles: there will be physical evaporation but also formation of azeotropic mixtures which will lower the temperature of evaporation. Additionally there are also thermo-oxidative degradation reactions since the drying process is performed in an air flow. The influence of the drying process on the outcome of VOCs should discussed in more detail. Tables should be revisited and commented.

Line 216: Rodriguez et al. [23]

Table 4: 0.5 ±0.2 should be shifted

Line 254: vapor pressure should be considered rather than evaporation temperature. Moreover some of the identified compound can form azeotropic mixture lowering the boiling point ; e.g. furfuryl alcohol/H2O (6°C instead of 98.5°C) toluene/ H2O (77°C instead of 111°C). Additionally the “drying” process is dynamic since there an air flow shifting continuously the equilibria. Tables should be revisited and commented.

Line 294: oxidative processes product of the decomposition reactions

Lines 293-310: in situ degradation during the drying process itself should also be commented and a distinction should be made between emission as the result of chemical and microbiological degradation during storage in situ emission due to the drying process itself.

Line 316 table 5: there is a conflict between table 5 and table 2 Phenol is now classified within aromatics while in table 3 it is considered as alcohol. As mentioned earlier it should be classified in within the class of phenolics for obvious reasons. The same for acetic acid which is an organic acid and not aromatic. Toluene is an aromatic hydrocarbon.

Lines 331-342: I refer again to my remark of different degradation process during storage and because of the drying process Since unsaturated hydrocarbons, esters and acids are detected these compounds can also be considered as precursors for in situ oxidation; Acetic acid is in my opinion a degradation product of hemicellulose and is most probably found during the drying process. Acetic acid, acetaldehyde, furan methanol and furfural are typical decomposition products from hemicellulose (thermal degradation of hemicellulose occurs within the temperature rage 200-380°C, cellulose within 320-400°C).

Reviewer 2 Report

The manuscript gives a comprehensive overview on a study on the odorant changes during thermal-mechanical drying of olive waste. The results are pre4sented in an appropriate way and the highlights are adequate. Therefore I recommend publishing the manuscript after minor revision with regard to the comments made below.

General Comment:

-          The English needs some improvement. E.g. Highlights, 2nd point better: „VOC concentrations from orujo and alperujo increase over time“

Specific comments:

L 52 ff:   Slightly more information on orujo and alperujo should be given. Here ist only mentioned, that they derive from a 2 stage and 3 stage processes. This is quite vague and leaves the reader alone with the probable differences of both types.

L 65:      Are the storage reservoirs not shielded against rain?

L 99 ff:   An entire paragraph on the influence of environmental conditions should at least – even with the references mentioned – include a brief comment on what the influences are.

Fig. 3      Please add Monthly (July – December) concentrations …

Author Response

Round  2

Reviewer 1 Report

The authors have answered remarks and comments of the reviewers and have modified the manuscript accordingly. Therefore the publication should be accepted in the present form.